# Clove Polyphenolic Compounds Improve the Microbiological Status, Lipid Stability, and Sensory Attributes of Beef Burgers during Cold Storage

**DOI:** 10.3390/antiox11071354

**Published:** 2022-07-12

**Authors:** Isam A. Mohamed Ahmed, Elfadil E. Babiker, Fahad Y. Al-Juhaimi, Alaa El-Din Ahmed Bekhit

**Affiliations:** 1Department of Food Science and Nutrition, College of Food and Agricultural Sciences, King Saud University, P.O. Box 2460, Riyadh 11451, Saudi Arabia; iali@ksu.edu.sa (I.A.M.A.); ebabiker.c@ksu.edu.sa (E.E.B.); faljuhaimi@ksu.edu.sa (F.Y.A.-J.); 2Department of Food Sciences, University of Otago, Dunedin 9054, New Zealand

**Keywords:** clove powder extract, antioxidant, antimicrobial, beef burger, quality attributes

## Abstract

This study investigated the phenolic composition of clove powder extract (CPE), determined by high-pressure liquid chromatography, as well as the effect of the clove powder (CP) concentration (0, 2, 4, and 6%) on the quality of beef burgers during 21 days of cold storage at 4 °C. The CPE contained a high amount of total phenolic content (455.8 mg Gallic acid equivalent/g) and total flavonoid content (100.4 mg catechin equivalent/g), and it exhibited high DPPH antioxidant scavenging activity (83.9%). Gallic acid, catechol, and protocatechuic acid were the highest phenolic acids (762.6, 635.8, and 544.9 mg/100 g, respectively), and quercetin and catechin were the highest flavonoid acids (1703.1 and 1065.1 mg/100 g, respectively). Additionally, the CPE inhibited the growth of both Gram-positive and Gram-negative bacteria effectively at 100 μg/disc. The addition of the CP had no discernible influence on the pH of the meat patties. The addition of CP at 4 and 6% increased the phenolic content and antioxidant activity of the beef patties, which consequently resulted in reduced lipid oxidation and microbial spoilage throughout the storage period. Furthermore, the CP significantly (*p* ≤ 0.05) improved the beef burger cooking characteristics (cooking yield, fat retention, moisture retention, and shrinkage). Additionally, the sensory acceptability was higher (*p* ≤ 0.05) for the burgers that contained 2% and 4% CP compared with the other treatments. In conclusion, the bioactive compounds in CP can extend the shelf life and improve the safety of beef burgers.

## 1. Introduction

Despite the current interest in plant-based foods, the production and consumption of meat products such as beef burgers have expanded dramatically due to the growth of the fast-food industry, increased demand, and the high nutritional value of meat products, with significant health benefits [1]. The oxidation of meat and meat products can shorten their shelf life and economic potential by altering the composition of muscle lipids and proteins, which affects the product’s quality. The meat industry has traditionally utilized synthetic antioxidants, such as butylated-hydroxyl-anisole (BHA) and butylated-hydroxyl-toluene (BHT), to control the oxidation in processed beef [2], but, in recent years, natural antioxidants or antioxidant-containing herbs and species have been of interest [3], and especially those rich in polyphenols [4]. Consumers and health experts are concerned about the use and safety of the aforementioned synthetic antioxidants [5], which has intensified the research on the beneficial properties of natural antioxidants for food preservation.

Many researchers [6,7,8] have studied the potential of natural antioxidants to prevent fat oxidation in different food products. Plant materials, such as spices and herbs, are rich in bioactive compounds, such as phenolics and flavonoids, which exhibit antioxidant and antibacterial properties [6]. For example, antioxidants derived from plants, such as *Moringa oleifera* seed powder [9], olive cake powder [1], pistachio seed hull extract [10], Argel leaf powder and extract [11,12], *Acacia nilotica* fruit and seed extracts [13,14], *Ephedra alata* extract [15], baobab (*Adansonia digitata*) seed extract [16], red pitaya extract [17]), and Colombian berry (*Vaccinium meridionale* Sw.) extracts [18], have been found to maintain the quality properties and enhance the storage stability of meat products.

Clove (*Syzygium aromaticum*) bud is the most important spice that is frequently used as a flavoring agent in the traditional and industrial food industries [19]. It is also used for numerous medicinal and pharmaceutical applications to prevent aging, promote wound healing, and treat many diseases, such as thyroid dysfunction, skin cancer, digestion problems, and cardiovascular diseases [20]. This is due to the fact that cloves have several bioactive phenolic compounds that exhibit high antioxidant, antiviral, antimicrobial, and anticancer activities [20,21]. Due to the high bioactive properties of clove, its application as a natural antioxidant in food products has greatly increased over the past decades [22]. The use of clove extract appears to influence the microbiological dynamics of food systems. For example, the addition of clove extract to kimchi paste considerably reduced the aerobic and lactic acid bacteria population, which postponed changes in the O_2_ and CO_2_ concentrations, pH, titratable acidity, and sugar content during storage [23]. In addition, the incorporation of clove extract and essential oil into meat products was found to improve the storage stability of the products during cold and frozen storage [22,24,25,26,27]. To the best of our knowledge, all studies on the application of clove in meat products were conducted using clove aqueous or alcoholic extracts. The use of clove extract carries an extra cost to prepare the extracts, and it also limits the shelf life of the active compounds, as it is more vulnerable to oxidation after its release from the parent matrix. However, studies on the utilization of clove powder (CP) as a natural antioxidant and preservative are scarce. Therefore, the current study was conducted to comprehensively investigate the phenolic components, total phenolic contents, and antioxidant and antibacterial activities of CP, as well as the effects of different concentrations (0, 2, 4, and 6%) of CP on the physicochemical, microbiological, and sensory-quality attributes of a beef burger during cold storage (4 °C).

## 2. Materials and Methods

### 2.1. Materials

Beef samples (boneless rounds) and beef back fat from young Holstein Friesians (*Bos taurus*) were obtained 24 h postmortem from a local market in Riyadh, Saudi Arabia. Chickpea powder, salt, garlic powder, white and black pepper, vinegar, onion powder, and cloves were purchased from the local supermarket. All chemicals and media used in this study were of analytical grade.

### 2.2. Preparation of Clove Powder (CP) and Extract (CPE)

Extraneous materials, such as leaves and soil, were removed from the cloves (obtained from local market, Riyadh, Saudi Arabia). Then, the cloves were washed with clean water, dried in a freeze-drier, milled, and sieved through a 1 mm sieve. The obtained CP was placed in plastic bags and stored at 4 °C for subsequent analyses and the preparation of beef burgers. Based on a preliminary investigation, the CP extracts (CPEs) were prepared by mixing CP with 80% methanol (at a ratio of 1:2 (*w*/*v*)). The use of 80% methanol was shown to produce the highest antioxidant activity compared with ethanol, methanol, and acetone at 80 and 100% concentrations [28]. The mixtures were stirred on a magnetic stirrer (Fisher, 14–511-1A, Tuscaloosa, AL, USA) for six h, and were sterilized by filtration using a 0.45 μm membrane filter (Millipore, Bedford, MA, USA). The filtrates were freeze-dried using a Labconco 8811 freeze-dryer (Labconco Corporation, Kansas, MO, USA), and the extracts were stored at −20 °C for subsequent analyses.

### 2.3. Characterization of CP Polyphenol Compounds Using HPLC

The phenolic compounds of CP were determined using a Shimadzu HPLC system equipped with a photometric-diode-array (PDA) detector, as described previously [29]. Briefly, 20 μL CPE was injected into an Inertsil ODS-3 (4.6 mm × 250 mm; 5 μm ID) and was eluted using a gradient of buffer A (0.05% acetic acid) and buffer B (acetonitrile) at a 1 mL/min flow rate for a total run of 1 h at 30 °C, and peaks were detected at 330 and 280 nm. External authentic standards of the phenolic compounds were used for the identification of the sample phenolic compounds, the areas under the peak were used for the quantification of the phenolic compounds, and the results were specified as mg/100 g sample.

### 2.4. Determination of the Antimicrobial Activity of CPE

The antimicrobial activity of the CPE was determined using the disc diffusion method, as described in the literature [30]. Pure strain cultures of *Bacillus cereus* ATCC 14579, *Bacillus coagulans* (laboratory isolate), *Clostridium perfringens* ATCC 13124, *Escherichia coli* ATCC 10536, *Klebsiella pneumonia* ATCC 10031, *Listeria monocytogenes* ATCC 19114, *Micrococcus luteus* ATCC 10240, *Pseudomonas aeruginosa* ATCC 9027, *Salmonella typhimurium* ATCC 14028, *Serratia marcescens* ATCC 13880, *Staphylococcus aureus* ATCC 29737, and *Yersinia enterocolitica* ATCC 27729 were used. The strains were cultured overnight in appropriate media, were adjusted to a final concentration of 10^6^ CFU/mL, and 1 mL was plated on nutrient agar media and was left for 20 min to dry. Then, the discs enriched with CPE (100 μg/disc) and penicillin (100 μg/disc) were carefully placed on the surfaces of the plates. Negative controls were prepared by using discs enriched with sterilized distilled water. After 24 h incubation at 37 °C, the inhibition zones around the discs were recorded.

### 2.5. Preparation of Beef Burgers

The method described by Hawashin et al. [1] was followed to prepare the beef burgers, using CP instead of CPE, as it is more economical and applicable for the industry. The formulation consisted of 4 concentrations of CP (0, 2, 4, and 6%), cold water (10.4%), minced beef meat (76, 74, 72, and 70%), added fat (10%), chickpea powder (2%), salt (1%), black pepper (0.2%), white pepper (0.2%), garlic powder (0.2%), and onion powder (2%) to obtain four levels of CP-formulated beef burgers. A Stephan mixer (Stephan U. Sohner UM 12 GmbH and Co., Gackenbach, Germany) was used to obtain a homogenous meat mixture and ingredients. The beef burgers (100 g each) were formed using a burger-making machine (Expro. Co., Shanghai, China). The formed beef burgers were placed individually in polyethylene bags and were stored at 4 °C for 21 days. The stored burgers were analyzed at 7-day intervals.

### 2.6. Preparation of Beef Burger Extract

Beef burger extracts were prepared by homogenizing 3 g of freeze-dried burger samples in 6 mL of 80% methanol at 10,000 rpm (T18 homogenizer, IKA, Wilmington, NC, USA) for 1 min. After 10 min centrifugation at 10,000× *g*, the mixtures were filtered through a 0.45 μm membrane filter and kept at −20 for further analysis.

### 2.7. Determination of Total Phenolic Content

The total phenolic content of the CP and burgers was determined using the Folin–Ciocalteu-reagent method, as described in the literature [31,32]. Gallic acid was used as a standard, and the results are expressed in mg gallic acid equivalent (GAE)/g.

### 2.8. Determination of Total Flavonoid Content

The total flavonoid content (TFC) of the CP was determined as described by Kim et al. [32]. Extract samples (1 mL) were mixed with distilled water (4 mL), 5% sodium nitrite solution (0.3 mL), and 10% aluminum chloride solution (0.3 mL). The mixture was kept at 25 °C for 5 min. Then, 1 M NaOH (2 mL) and distilled water (10 mL) were added to the mixture, and the absorbance was read at 510 nm using a spectrophotometer (Model UV 2005; Selecta, Barcelona, Spain; 22). Catechin was used to contrast a standard curve, which was used to calculate the TFC. The results were presented in mg catechin equivalents/g (mg CE/g sample).

### 2.9. Determination of Antioxidant Activity

The antioxidant activity of the CP and burger samples was determined using the 2,2-diphenyl-1-picrylhydrazyl radical scavenging activity (DPPH scavenging activity), as described by Li et al. [33]. Briefly, 1 mL of the extract was mixed with 2 mL of 0.25 mmol/L DPPH solution, vortex-mixed, and allowed to stand for 10 min in the dark at room temperature (25 °C). Methanol was used as a control, and the absorbance of the mixtures was read with a spectrophotometer (Lambda EZ 150 spectrophotometer, PerkinElmer, Waltham, MA, USA) at 518 nm. The DPPH scavenging activity was determined using the following formula [33]:$$\mathrm{DPPH}\;\left(\%\right)=\left(\frac{A_{\mathrm{control}}-A_{\mathrm{sample}}}{A_{\mathrm{control}}}\right)\times 100$$
where *A* is the absorbance recorded, and DPPH is expressed as percentage inhibition.

### 2.10. pH Measurement

The pH was measured using a pH meter probe (Corning Scientific Products, Corning, NY, USA), as described previously [13]. A total of 1 g of the freeze-dried burger was homogenized in 10 mL distilled water using an ultraturrax T18 homogenizer (IKA, Wilmington, NC, USA), and the pH was measured directly.

### 2.11. Cooking of the Beef Burgers

Cooking of the burgers was performed using a conventional oven (Hobart Corp., Troy, OH, USA) at 180 °C, and the burgers were rotated every 10 min to ensure even heat distribution. A digital probe thermometer (Oakton, Eutech Instruments, Shanghai, China) was used to monitor the cooking of the burgers until the center of the sample reached 80 °C. 

### 2.12. Evaluation of Cooking Properties

The cooking properties (cooking yield %, dimensional shrinkage %, fat %, and moisture retention %) of the burgers were evaluated as described previously [1], using the following equations:$$Cooking\;loss\;\left(\%\right)=\frac{Cooked\;weight}{Raw\;weight}\times 100$$
$$Fat\;retention\;\left(\%\right)=\frac{Cooked\;weight\times percent\;fat\;in\;cooked\;patties}{Raw\;weight\times percent\;fat\;in\;raw\;patties}\times 100$$
$$Moisture\;retention\;\left(\%\right)=\frac{Percent\;yield\times percent\;moisture\;in\;cooked\;patties}{100}$$
$$Shrinkage\;\left(\%\right)=\frac{\left(Raw\;pattie\;thickness-Cooked\;pattie\;thickness\right)+\left(Raw\;pattie\;thickness-Cooked\;pattie\;thickness\right)}{Raw\;pattie\;thickness+Raw\;pattie\;diameter}\times 100$$

### 2.13. Microbiological Analysis

The total-plate-count method of the International Organization for Standardization (ISO 4833-1) [34] was used to evaluate the raw beef burgers during the cold-storage period. Briefly, 10 g of burger samples were subjected to 10 min homogenization in 90 mL of sterilized saline solution (0.85% NaCl). Then, 10-fold serial dilutions were prepared by diluting 1 mL of the sample homogenate with 9 mL of sterile saline solution, and suitable dilutions were pour-plated into the plate count agar (Oxoid, Hampshire, UK). The plates were incubated at 30 °C for 48 h, and the colonies were counted and expressed as log10 CFU/g.

### 2.14. Determination of Thiobarbituric Acid Reactive Substances (TBARS) 

The TBARS values of the raw burger samples were determined as described previously [14]. Briefly, burger samples (5 g) were homogenized in distilled water (20 mL) and were filtered using filter paper (Whatman No. 1). Thereafter, the filtrate (1 mL) was added to 4 mL thiobarbituric acid (20 g/mL) and 100 µL butylated hydroxytoluene (10 g/100 mL) in a screw-cap tube. The sample was mixed vigorously and heated for 10 min in a water bath (95–100 °C) to facilitate the development of pink color. Then, the sample was centrifuged (5500× *g*) for 25 min, and the absorbance of the supernatant was read at 532 nm. A standard curve was prepared using the standard 1,1,3,3-tetraethoxypropane, and the results were expressed as mg malonaldehyde/kg sample.

### 2.15. Measurement of Color Attributes

The color of the raw burger samples was measured using a Hunter Lab colorimeter (Miniscan^®^ XE plus 4500L, Reston, VA, USA) fitted with a 25 mm aperture and calibrated using Illuminant D-65 and 10° observer. The colorimeter was calibrated using black and white standard tiles (X = 80.4, Y = 85.3, Z = 91.5), according to the manufacturer’s guidelines. The color coordinates (L*, a*, and b*, corresponding to lightness, redness, and yellowness, respectively) were read off the machine, and the color intensity (Chroma (C)) was calculated using the equation C = (a^2^ + b^2^)^0.5^.

### 2.16. Sensory Evaluation

An in-house trained panel of 20 members (20–35-year-old male staff and students at the College of Food and Agriculture, King Saud University) evaluated the sensory characteristics of the cooked burger samples using a 9-point hedonic scale. Preliminary training sessions (*n* = 3) were provided at the panelists’ prior evaluation to acquaint them with the sensory attributes that we expected to measure. The samples were coded with three-digit random numbers and were randomly presented to the panelists. The panelists were asked to evaluate the color, flavor, texture, taste, juiciness, and overall acceptability on a 0-to-9 scale, with the 0 anchors labeled dislike extremely, and the 9 anchors labeled as like extremely. The sensory evaluation was carried out in a sensory evaluation room and was performed in three sessions at each storage time (0, 7, 14, and 21 days). The mean values of the scores for twenty participants were calculated for each sample and session and were used for data analysis. 

### 2.17. Statistical Analysis

The experiments were designed by applying a completely randomized block design with four treatments (control, 2% CP, 4% CP, and 6% CP), and determinations were taken on four storage days (0, 7, 14, and 21 days). The entire blocks were independently repeated three times. Analysis of variance (ANOVA) and Duncan’s multiple range test (SAS 8.0 software, SAS Institute, Inc., Cary, NC, USA) were applied to determine the effects of treatments and storage times, as well as their interaction, on the physicochemical, microbial, and sensory properties of the beef burgers. In the models, the treatments, storage periods, and their interaction were assigned as the fixed effects, and the replications of the experiments were the random effects. The scores of the different sensory characteristics were compared between treatments and storage periods using the general linear model (GLM). Duncan’s multiple range test was used to compare the means to determine the effects of treatments and storage. Data were expressed as mean and standard error (SE), and statistical significances of the experimental data were accepted at a probability of *p* ≤ 0.05. MULTBIPLOT software was used for the principal component analysis (PCA) and hierarchical cluster analysis (HCA) of the data variables of the samples during the storage intervals, as described by Vicente-Villardón [35].

## 3. Results and Discussion

### 3.1. Phenolic Compounds, Total Phenolic Content, Flavonoid Content, and Antioxidant Activity of CPE

The individual phenolic compounds, total phenolic content (TPC), total flavonoid content (TFC), antioxidant activity, and antimicrobial activity of the CPE are shown in Table 1. The data showed that the CPE had a high amount of TPC (455.8 mg GAE/g), TFC (100.3 mg CE/g), and antioxidant activity (83.9%). Some studies have evaluated the bioactive properties of extracts obtained from different cloves using different extraction solvents and conditions. Ali et al. [36] extracted bioactive compounds from CP using 70% ethanol in MilliQ water containing 0–1% formic acid, and they reported TPC and TFC contents of 215.1 mg GAE/g and 5.6 mg quercetin equivalents (QE)/g, respectively. Frohlich et al. [37] applied an ultrasound-assisted extraction method using aqueous ethanol (70% at 60 °C for 20 min) as a solvent for extracting bioactive compounds from clove leaves and found a TPC of 387.9 mg GAE/g. El-Maati et al. [19] used three different solvents (water, ethanol, and ethyl acetate) for the extraction of phenolic compounds from clove bud, and they observed that water is the best solvent, yielding TPC, TFC, and DPPH radical scavenging values of 230 mg GAE/g extract, 17.5 mg QE/g extract, and 91.4%, respectively. Sultana et al. [38] extracted bioactive compounds from clove seeds using different solvents (30% and 70% aqueous methanol, and 0.5–1.0 N HCl acidified methanol) for 24 h and found that acidified methanol extracts contain high TPC (115.3 mg GAE/100 g dry matter) and DPPH radical scavenging activity (94.6%). Suantawee et al. [39] extracted clove buds using hot water (95 °C) for 3 h and obtained an extract that had TPC and TFC values of 239.6 mg GAE/g extract and 65.7 mg CE/g extract, respectively. The results of the TPC, TFC, and DPPH inhibition in the aforementioned studies varied compared with those found in the present study. The variations in the bioactive properties (TPC, TFC, and DPPH inhibition) among these studies could be attributed to the differences in the genotypes, parts, maturity stages, cultivation locations and seasons, the postharvest processing conditions of the cloves, the extraction conditions (extraction method, solvent type, polarity, sample/solvent ratio, temperature, and duration) of the bioactive compounds, as well as the reference standards used in the assays. 

The phenolic profile of the CPE showed fourteen phenolic compounds, classified into phenolic acids of hydroxybenzoic (catechol and gallic, protocatechuic, and syringic acids) and hydroxycinnamic (caffeic, p-coumaric, trans-cinnamic, and trans-ferulic acids) acids, flavonoids (namely flava-3-nols [(+)-catechin]), flavonols (quercetin, isorhamnetin, and rutin trihydrate), flavones (apigenin 7 glucoside), and stilbenes (resveratrol) (Table 1).

Among these groups, hydroxybenzoic acids and flavonols were abundant, with total values of 2081 mg/100 g and 1927.3 mg/100 g, respectively, followed by flava-3-nols (1065.1 ± 7.62 mg/100 g), hydroxycinnamic acids (270.4 mg/100 g), flavones (99.7 mg/100 g), and stilbenes (19.0 mg/100 g). The most abundant phenolic compounds of the CP were quercetin (1703.1 mg/100 g) and (+)-catechin (1065.1 mg/100 g), followed by gallic acid (762.6 mg/100 g), catechol (635.8 mg/100 g), and protocatechuic acid (544.9 mg/100 g), whereas the least abundant was trans-cinnamic acid (0.78 mg/100 g). These findings suggest the high abundance of phenolic compounds in clove bud powder. Previous studies have evaluated the phenolic profile of clove bud extracts using different extraction conditions and chromatographic analysis means. Hassan et al. [40] analyzed clove water extract using HPLC and found that rutin (7588.0 mg/100 g) was the major phenolic compound, followed by benzoic acid (106.2 mg/100 g), ellagic acid (100.8 mg/100 g), myricetin (77.7 mg/100 g), and quercetin (68.3 mg/100 g), whereas the least abundant were caffeic (2.03 mg/100 g) and p-coumaric acid (2.26 mg/100 g). Adefegha et al. [41] analyzed the water extract of clove bud using HPLC–DAD and reported quercitrin (3026 mg/100 g) and quercetin (2971 mg/100) as the most abundant phenolic compounds, followed by kaempferol, chlorogenic acid, gallic acid, caffeic acid, luteolin, and rutin, whereas the least abundant were catechin and ellagic acid. Ali et al. [36] analyzed the aqueous extract of CP using the LC–ESI–QTOF–MS and HPLC–PDA methods, and they found an abundance of phenolic acids, namely, p-coumaric, sinapic, chlorogenic, and trans-ferulic acids. Shan [42] analyzed the phenolic compounds in an aqueous methanolic extract (80%, 24 h, 23 °C) of CP using HPLC–DAD and found high amounts of gallic acid (2375.8 mg/100 g) and phenolic acids (namely, caffeic, ferulic, and ellagic acids), as well as flavonoids (namely, quercetin and kaempferol). The discrepancies in the phenolic compounds among these studies are likely due to the differences in the clove type, maturity stages, cultivation locations and seasons, the postharvest processing conditions, the extraction conditions (extraction method, solvent type, temperature, and duration), and the identification and quantification methods of the phenolic compounds. The above results indicated that CP is rich in bioactive compounds and consequently can be utilized as a functional ingredient to delay oxidative processes during extended storage periods of food products. The major phenolic compounds (quercetin, (+)-catechin, gallic acid, catechol, and protocatechuic acid) identified in this study have high nutritional and health potential due to their wide range of biological and pharmaceutical activities [43].

### 3.2. Antimicrobial Activity of CP Extract (CPE)

The effects of the CP extract (100 μg/disc) on twelve Gram-negative and Gram-positive bacteria compared with penicillin as a standard antibiotic are shown in Table 1. The inhibition-zone values of the CPE for *Pseudomonas aeruginosa* ATCC 9027 and *Clostridium perfringens* ATCC 13124 were comparable to that of penicillin, whereas those for *Escherichia coli* ATCC 10536, *Serratia marcescens* ATCC 13880, *Salmonella typhimurium* ATCC 14028, *Yersinia enterocolitica* ATCC 27729, *Listeria monocytogenes* ATCC 19114, *Bacillus coagulans* (laboratory isolate), and *Staphylococcus aureus* ATCC 29737 were lower compared with penicillin. Furthermore, the inhibition-zone values of the CPE for *Klebsiella pneumoniae* ATCC 10031, *Bacillus cereus* ATCC 14579, and *Micrococcus luteus* ATCC 10240 were significantly (*p* ≤ 0.05) higher than that of penicillin. These findings are comparable to those reported previously for different clove extracts by many researchers. El-Maati et al. [19] observed that clove extract at a 100 μg/mL concentration possessed high antimicrobial activity (inhibition zones of 10–17 mm) against *E. coli* ATCC 8739, *L. monocytogenes* Scott A, *S. enteritidis* PT4, *S. marcescens*, and *S. aureus* ATCC 6538. Singh [44] reported that clove extract exhibited higher antimicrobial activity (17–25 mm inhibition zone) against *E. coli*, *L. monocytogenes*, *Pseudomonas aeruginosa*, *S. aureus*, and *S. epidermidis* compared with the antibiotic Cefotaxime (8–10 mm inhibition zone). Rosarior [45] stated that 2000 μg of clove ethanolic extract showed broad-spectrum inhibition (12.3–19.7 mm inhibition zone) against *E. coli*, *K. pneumonia*, *Proteus mirabilis*, *S. aureus*, and *S. epidermidis*. The antimicrobial activity of clove extract in the present study could be attributed to phenolic compounds such as quercetin, (+)-catechin, gallic acid, catechol, and protocatechuic acid [46], which are present in high quantities in the extract. In agreement with our findings that Gram-positive strains are more susceptible to clove extract than Gram-negative strains, Shan et al. [42] and El-Maati et al. [19] reported that Gram-positive bacteria were more susceptible to antibacterial agents than Gram-negative bacteria due to the structure of the outer layer of the cell membrane, which consists of dense lipopolysaccharide molecules, and perivascular enzymes in Gram-negative bacteria. Therefore, antimicrobial compounds can easily disintegrate the cell walls and cytoplasmic membranes of some Gram-positive bacteria and release the contents of the cytoplasm [42], which eventually causes their death. Our study revealed that the CPE is a natural antimicrobial agent that could play a role in extending the shelf lives of food products by controlling the growth of microorganisms.

### 3.3. Cooking Properties of Burger–CP Formulate during Cold Storage

The beef burger formulation with different concentrations of CP had a significant (*p* ≤ 0.05) effect on the cooking properties of the burgers (Table 2). At the beginning of storage, the incorporation of CP into the burger formulation significantly (*p* ≤ 0.05) increased the cooking yield (CY), fat retention (FR), and moisture retention (MR) to the maximum values at a 2% concentration, which slightly reduced as the concentration of CP was elevated to 4% and 6%. At 7, 14, and 21 days of storage, the highest FR values were observed in the burgers containing 2%, whereas the CY and MR values were highest in the patties containing 6% CP at 7 and 14 days of storage, and 2% CP stored for 21 days. The control beef burgers had significantly higher (*p* ≤ 0.05) dimensional shrinkages (DSs) than the CP-containing beef burgers (Table 2). At all storage times, increasing the concentration of the CP in the burger significantly (*p* ≤ 0.05) decreased the dimensional shrinkages of the beef burgers during the cooking process. The shrinkage of burgers during cooking is affected by many factors, including the loss of melted fat and juices, muscle-protein denaturation, and water evaporation, which are expected to affect the texture of cooked burgers. The findings of this study are comparable to those of meat patties formulated with finger millet flour [47], destoned olive cake powder [1], and argel leaf powder [11]. The increased CY, MR, and FR and reduced DS in burgers containing CP are likely due to the ability of CP to modify the matrix structure of beef burgers and thereby increase the fat- and moisture-retention capabilities of the product. The ability of CP to retain more water and fat within the burger matrix could increase the cooking yield of the formulated product [11]. In addition, the binding ability of CP could stabilize the burger-matrix structure, reduce the losses of moisture and juiciness, and thereby retain the shape and size of the burger [47]. Moreover, retaining more moisture and fat within the burger matrix could improve the physical properties and sensory-quality attributes of the product [48]. In the current study, all of the clove bud powder was used, which might contain fibers, protein, and starch, in addition to the phenolic compounds. The interaction of these components with the burger matrix during the formulation and subsequent cooking could lead to the modification of the matrix structure and thereby improve the cooking properties of the burger.

### 3.4. pH of Raw Burgers during Cold Storage

As shown in Table 3, no significant effect of the CP addition on the burger’s pH was observed throughout the storage period (21 days), compared with the control samples. A slight decrease in the pH level occurred during cold storage, which could be attributed to acid produced by acid-producing bacteria [49]. The present study showed that the addition of CP stabilized the pH of the beef burger for 14 days for all the beef burgers containing 2, 4, and 6% CP. The findings of this study are in agreement with those reported for beef patties formulated with 0.1% clove extract during cold storage for 10 days, where no significant changes in the pH values of the patties were observed [25]. In addition, no significant changes in the pH were observed in clove-extract-containing beef patties after 7 days [24] and 10 days [26] of cold storage. However, during frozen storage (−20 °C) for up to 6 months, the pH values of beef patties containing 0.1% clove extract significantly increased due to the formation of ammonia from amino acids during protein denaturation [27].

### 3.5. Microbial Analysis and Antioxidant Activity of Raw Burgers during Cold Storage

At 0 days of storage, the untreated control burger samples had a significantly (*p* ≤ 0.05) higher plate count than the CP-containing burger samples. It was observed that, as the concentrations of CP increased in the treated burgers, the plate count was significantly (*p* ≤ 0.05) decreased. As the storage period increased, there was an increase in the plate count in both the treated and control beef burgers; however, the increase in the control samples was significantly (*p* ≤ 0.05) higher than in the CP-treated samples over the storage period. The plate count of the control burger was greater than 7.0 log CFU/g on days 14 and 21 of storage, which rendered the burger unsafe for consumption; therefore, further storage and analyses were not performed for these burger samples. The lowest plate count was observed in the burger treated with 6% CP at the end of the storage period. The significant reduction in microbial growth observed in the CP-containing beef burgers could be attributed to the antimicrobial activity of the CP (Table 1), and thus CP can be considered a green processing preservative that can improve the storage stability of burgers. Plant materials rich in TPC, such as the CP, can preserve meat and meat products and improve the stability of the product during storage [1,9,10,11,22]. This was confirmed in the present study by investigating the total phenolic content (TPC) in the treated beef burgers. Increasing the CP content in the burger significantly (*p* ≤ 0.05) increased the TPC in the CP-treated burgers (Table 3). During storage, the TPC of CP-containing burgers is reduced due to the hydrolysis and utilization of phenolic compounds to prevent the peroxidation of the burger [14]. Phenolic-rich plants and their extracts have been gaining much interest due to their biological activities, including antimicrobial and antiviral activities, as well as antioxidant activities [50,51,52], which are greatly affected by the contents and type of individual polyphenols.

The increase in the TPC content in the treated beef burgers also led to higher antioxidant activity in the burger samples. During storage, the antioxidant activity of all the CP-containing burgers concomitantly reduced as the storage time progressed. These findings are in agreement with previous studies from our research groups that demonstrated that the incorporation of plant extracts into meat burgers caused a significant increase in the TPC and antioxidant activity of the burgers [10,14]. It should be noted here that the TPC and antioxidant-activity analyses were not determined in the control samples on 14 and 21 days of cold storage due to the high microbial load, which led us to consider them unsafe for human consumption.

As shown in Table 3, the lipid peroxidation (TBARS) in the beef burgers was significantly reduced (*p* ≤ 0.05) with the incorporation of CP compared with the untreated burger samples. An increase in the level of CP in the burger was accompanied by a reduction (*p* ≤ 0.05) in the level of TBARS, with the lowest value observed in beef burgers treated with 6% CP. Among the four treatments, the TBARS increased with the increasing storage period, which indicated the continued aldehyde production in the stored products. However, the burgers treated with 6% CP had the lowest TBARS value (0.82 mg malonaldehyde/kg sample) on day 14, compared with the other treatments at the same storage period. The lipid oxidation was reduced by the inclusion of CP in the beef burgers due to the phenolics present in the CP. This is in agreement with findings observed in meat patties containing clove extract [25,26,27], argel leaf water extract [12], baobab seed extract [16], *Acacia nilotica* seed extracts [14], and Moringa seed powder [9]. The presence of antioxidant compounds in the treated burgers is expected to control the oxidative processes in the beef burgers during the cold-storage period. A reduction in the TPC and antioxidant activity in the treated burgers was observed, which suggests the consumption of these activities in defending against the oxidative process during the storage period. A similar observation in pork burgers containing moringa leaf extract [53], chicken burgers containing *Acacia nilotica* seed extract [14], and chicken meat treated with clove extract [22] has been reported during the cold storage of these products. The high TPC and antioxidant activity in the CP-containing burgers during the storage indicated that the incorporation of the CP was successful at controlling the oxidation of lipids and prevented any undesirable reaction that may have had harmful effects on the product during storage, such as microbial growth, thereby improving the health benefits and extending the shelf life of the burger.
antioxidants-11-01354-t003_Table 3Table 3pH, plate count, and oxidative characteristics of raw beef burger formulated with different concentrations of clove powder during storage at 4 °C (±1).Storage Period (Days)Clove Powder (%)0246pH05.14 ± 0.465.01 ± 0.315.04 ± 0.595.11 ± 0.3375.41 ± 0.275.32 ± 0.455.67 ± 0.365.82 ± 0.4514ND5.32 ± 0.565.24 ± 0.455.42 ± 0.6721ND4.84 ± 0.394.69 ± 0.664.78 ± 0.72Plate count (log CFU/gm)03.36 ± 0.21 ^bp^
3.26 ± 0.19 ^cp^
2.88 ± 0.14 ^bq^
2.65 ± 0.37 ^bq^
76.61 ± 0.32 ^ap^
4.49 ± 0.18 ^bq^4.39 ± 0.31 ^aq^
3.03 ± 0.39 ^ar^
14ND5.65 ± 0.58 ^ap^4.60 ± 0.48 ^aq^
3.36 ± 0.25 ^ar^
21ND5.86 ± 0.67 ^ap^4.66 ± 0.59 ^aq^
3.53 ± 0.56 ^ar^
Total phenolic content (TPC) (mg GAE/g sample)01.52 ± 0.46 ^ar^4.63 ± 0.25 ^aq^4.79 ± 0.37 ^aq^6.49 ± 0.28 ^ap^71.21 ± 0.57 ^as^4.03 ± 0.03 ^br^4.61 ± 0.18 ^aq^5.44 ± 0.34 ^bp^14ND3.29 ± 0.19 ^cq^4.50 ± 0. 47 ^ap^4.57 ± 0.47 ^cp^21ND2.93 ± 0.11 ^dr^3.19 ± 0. 17 ^bq^4.24 ± 0.61 ^cp^DPPH (% inhibition)09.99 ± 0.57 ^bs^52.3 ± 0.68 ^ar^57.9 ± 0.27 ^aq^69.3 ± 0.26 ^ap^713.40 ± 0.88 ^as^30.6 ± 0.47 ^br^35.3 ± 0.47 ^cq^47.9 ± 0.47 ^cp^14ND23.5 ± 0.49 ^cr^19.9 ± 0.35 ^dq^50.2 ± 0.75 ^bp^21ND22.7 ± 0.26 ^dr^36.8 ± 0.44 ^bq^49.9 ± 0.35 ^bp^Thiobarbituric acid reactive substances (TBARS) in mg malonaldehyde/kg sample00.99 ± 0.04 ^ap^0.83 ± 0.05 ^cq^0.81 ± 0.05 ^bq^0.72 ± 0.01 ^cr^71.10 ± 0.11 ^ap^0.91 ± 0.02 ^bq^0.87 ± 0.05 ^bq^0.79 ± 0.03 ^br^14ND0.93 ± 0.04 ^bp^0.91 ± 0.02 ^bp^0.82 ± 0.05 ^bq^21ND1.15 ± 0.02 ^ap^1.07 ± 0.03 ^aq^0.99 ± 0.03 ^ar^Values are presented as means of triplicate samples (±SE). Means not sharing common superscript(s) a, b, c, or d in a column or p, q, r, or s in a row are significantly different at *p* ≤ 0.05, as assessed by Duncan’s multiple range test. ND: not determined (spoiled).

### 3.6. Color Characteristics of Raw Beef Burger–CP Formulate during Storage

Table 4 shows the color characteristics of the control and treated raw beef burger during cold storage at 4 °C for 21 days.

The incorporation of CP into the beef burgers significantly (*p* ≤ 0.05) improved the color properties (a*, b*, and c*) compared with the control at the beginning of the storage, and, among all the formulated burgers, the 4%-CP-containing burgers had the highest values of these color attributes. The L* values of the CP-containing burgers were lower than the control (*p* ≤ 0.05). The increases in the a*, b*, and c* of the beef burgers following the addition of the CP could be due to the color of the CP, which is brown. The reduction in the L* value of the beef burger following the incorporation of CP is likely due to the dilution of the burger color by the added CP. The use of plant materials can exert an effect regardless of the antioxidant activity because many of the plant materials and their extracts contain natural pigmented materials, such as polyphenols [54]. Therefore, the difference in the color parameters may be due to the pigments present in the plant material used, which may have a significant effect on the color of the burgers. The findings of this study are comparable to the reports indicating a reduction in the L* and increase in the a* and b* values of meat patties formulated with clove extract [25], grape tomato powder [55], destoned olive cake powder [1], and Salvia (Salvia officinalis) powder [56]. During storage, the a* values of all the CP-containing burgers increased (*p* ≤ 0.05) to the maximum value on day 7 of storage, and they reduced again as the storage period elongated. However, a concomitant (*p* ≤ 0.05) reduction in the L* values was observed in all the CP-containing burgers. In addition, the b* and c* values of the burgers formulated with 2 and 4% CP were reduced as the storage time progressed, whereas those of the 6%-CP-containing burgers were increased and reached maximum values at day 14 of storage, and then reduced at the end of storage (*p* ≤ 0.05). The reduction in the color attributes of the CP-containing beef burgers at the end of storage could be justified by the greater release of pigments during storage [56]. A decrease in the color properties during the storage period was reported in a burger formulated with destoned olive cake [1]. The increase in the a* value during storage was also reported in meat products treated with clove extract [22,25,26] and was attributed to the preventive effect of the clove extract on the discoloration of the beef patties, which could also justify the increased a* value in the CP-containing burger during the storage treatment of the current study. In addition, CP contains high bioactive compounds and possesses high antioxidant activity; consequently, it could have prevented lipid and protein oxidation in the beef burger and thereby contributed to the color stabilization of the burger.

### 3.7. Sensory Properties of Beef Burger—CP Formulation

Table 5 shows the sensory characteristics of the cooked beef burger treated with different CP concentrations during cold storage. As the storage period increased, the CP-containing beef burger samples at the highest concentration (6%) had significantly (*p* ≤ 0.05) lower color acceptability compared with the control and other CP-treated samples. This could be due to the slightly brown color of the CP, which might have led to the burger being unacceptable to the panelists. Mitsumoto et al. [57] reported similar observations, with a slight color change in beef and chicken burger incorporated with tea catechin. The addition of CP improved the flavor, taste, texture, and juiciness after cooking compared with the control burger samples (*p* < 0.05). Despite the fact that the organoleptic properties were decreased with the progression of the storage time, the treated burgers were still better than the control ones. This substantiates the findings reported by Hawashin et al. [1], who observed a slight decrease in the organoleptic properties of a beef burger prepared using destoned olive cake during cold storage. The burger prepared with 2% CP had the highest overall acceptability compared with other treatments at the end of the storage period. The present findings indicate that the incorporation of 2% to 4% CP into beef burger could extend the shelf life of the burger without affecting its organoleptic properties.

### 3.8. Chemometric Analysis

To assess the interactive effects of the treatments (0, 2, 4, and 6% CP) and storage times (0, 7, 14, and 21 days) on the cooking, color, bioactive properties, microbial load, and sensory attributes of the beef burger, principal component analysis (PCA) and hierarchical clustering analysis (HCA) were performed, and the results are depicted in Figure 1. The PCA results showed the excellent contribution of the first and second principal components (PC1: 51.67%; PC2: 31.07%) to the total variability (82.74%) of the data (Figure 1A). PC1 had the sensory attributes, color properties, and antioxidant characteristics as the primary components, whereas PC2 had the cooking properties as the main components. In the biplot, the correlations between the traits are verified by the cosine of the angle between the vectors of the traits, in which positive and negative correlations are indicated by acute (<90°) and obtuse (>90°) angles, respectively [58]. In this sense, positive correlations were observed among the sensory attributes, color properties, bioactive properties, pH, cooking yield, and moisture retention, which demonstrates the association among these attributes. The aforementioned traits correlated negatively with the TBARS and total plate count. In addition, a high positive correlation was evident between the TBARS and total plate count, which indicates the strong association between them, whereas a negative correlation was seen between the fat retention and dimensional shrinkage. Two distinct groups of treatments and storage times were seen, according to their influence on the physicochemical, microbial, and sensorial-quality attributes of the beef burgers. The first group (right of the graph, triangle symbol) is composed of the burger formulated with different concentrations (2, 4, and 6%) of CP and stored for 0 to 7 days. This group is characterized by higher levels of sensory attributes, color properties, cooking yield, moisture retention, bioactive properties (DPPH and TPC), and pH than the other group, which demonstrates the significance of CP and a storage time of up to 1 week on conserving the quality attributes of beef burgers. Within this group, the fresh burgers containing CP had higher sensory attributes and color characteristics (L*, b*, and C*) than the other samples, whereas those stored for 7 days had higher bioactive properties, a*, cooking yields, and moisture retentions than the others. The second group (left of the graph, diamond symbol) contained the control burger at 0 and 7 days of storage and the CP-containing burger (2, 4, and 6%) stored for longer times (14 and 21 days). This group is characterized by higher levels of TBARS, total plate counts, and dimensional shrinkages than the other treatments and storage durations. Within this group, the control burgers stored for 7 days had a higher TBARS value and total plate counts than the other samples. These findings indicate that the control burger samples are more susceptible to deterioration and could deteriorate during the first week of cold storage, whereas those formulated with CP are more stable and can be stored for a long time at 4 °C. HCA was also conducted to deeply observe the effects of the treatments and storage conditions on the quality attributes of the beef burgers. In the HCA heatmap, the red color indicates the highest interrelation values, whereas the green color indicates the lowest interrelation values (Figure 1B). Based on the phenotyping similarities of the quality attributes of the beef burgers, the major four horizontal axes separate the samples into two vertical branches. The upper branch (control beef burger stored for 0 and 7 days and CP-containing burger stored from 14 to 21 days) had high levels of TBARS, total plate count, dimensional shrinkage, b*, and Chroma. The lower branch (CP-containing burger stored for 0 and 7 days) had high values of the bioactive properties, sensory attributes, color properties, cooking yield, and moisture retention, and, within this branch, the highest values of all these traits are evident in the fresh beef burger containing 4% and 2% CP, followed by that containing 6% CP. Overall, the incorporation of CP into the beef burger formulations at concentrations of 2 and 4% greatly improved the quality attributes of the fresh and 7-day-stored beef burgers.

## 4. Conclusions

The study showed that CP has a high level of TPC and antioxidant activity. Incorporating 6% CP into beef burgers enhanced the antioxidant activity and microbial stability and reduced the fat oxidation. However, it slightly affected the organoleptic properties of the burgers compared with those made with 2% and 4% CP. Thus, a CP concentration from 2% to 4% can act as a functional ingredient in beef burgers, extending the storage period, improving the acceptability, and providing many health benefits to the consumer and the product.

## Figures and Tables

**Figure 1 antioxidants-11-01354-f001:**
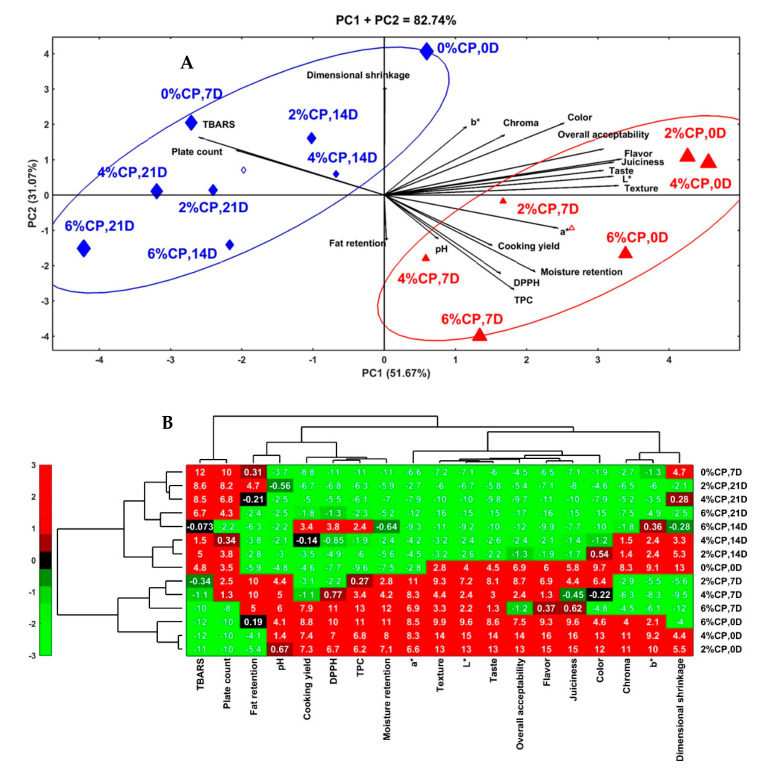
Principle component analysis (PCA) biplot (**A**) and hierarchical clustering analysis (HCA) heatmap (**B**) of the physicochemical and sensory-quality attributes of beef burger formulated with 0, 2, 4, and 6% clove powder, and stored at 4 °C for 0, 7, 14, and 21 days. In the heatmap, the red color indicates high linkage (positive relation), and green indicates low linkage (negative relation).

**Table 1 antioxidants-11-01354-t001:** Oxidative characteristics and antimicrobial activities of clove powder (CP) and clove powder extract (CPE).

Phenolic Compounds (mg/100 g)		
**Phenolic acids**	CP	CPE
**Hydroxybenzoic acids**		
Gallic acid	762.6 ± 3.65	63.6 ± 0.30
Protocatechuic acid	544.9 ± 2.80	45.4 ± 0.23
Syringic acid	137.7 ± 2.10	11.5 ± 0.18
Catechol	635.8 ± 3.45	53.0 ± 0.29
**Hydroxycinnamic acids**		
Caffeic acid	151.7 ± 0.56	12.6 ± 0.05
*p*-Coumaric acid	16.1 ± 0.39	1.3 ± 0.03
*trans*-Ferulic acid	101.8 ± 0.63	8.5 ± 0.05
*trans*-Cinnamic acid	0.78 ± 0.05	0.1 ± 0.002
**Flavonoids**		
*Flavan-3-ols*		
(+)-Catechin	1065.1 ± 7.62	88.8 ± 0.64
*Flavonols*		
Quercetin	1703.1 ± 5.26	141.9 ± 0.44
Rutin trihydrate	98.1 ± 1.12	8.2 ± 0.09
Isorhamnetin	126.0 ± 2.96	10.5 ± 0.25
*Flavones*		
Apigenin 7 glucoside	99.7 ± 1.25	8.3 ± 0.10
**Stilbenes**		
Resveratrol	19.0 ± 0.96	1.6 ± 0.08
**Antioxidant activity**		
DPPH (%)	-	83.9 ± 0.41
Total phenolic content (mg GAE/g)	-	455.8 ± 2.51
Total flavonoid content (mg CE/g)	-	100.4 ± 1.40
**Antimicrobial activity**		
Bacteria	Inhibition zone (mm)	
	CPE (100 μg/disc)	Penicillin (100 μg/disc)
*Escherichia coli* ATCC 10536	20.0 ^b^ ± 0.05	26.0 ^a^ ± 0.09
*Serratia marcescens* ATCC 13880	25.0 ^b^ ± 0.02	30.0 ^a^ ± 0.12
*Pseudomonas aeruginosa* ATCC 9027	20.0 ± 0.10	20.0 ± 0.19
*Salmonella typhimurium* ATCC 14028	18.0 ^b^ ± 0.05	19.0 ^a^ ± 0.13
*Yersinia enterocolitica* ATCC 27729	11.0 ^b^ ± 0.02	13.0^a^ ± 0.04
*Klebsiella pneumoniae* ATCC 10031	17.0 ^a^ ± 0.12	16.0 ^b^ ± 0.08
*Bacillus cereus* ATCC 14579	22.0 ^a^ ± 0.07	18.0 ^b^ ± 0.18
*Clostridium perfringens* ATCC 13124	16.0 ± 0.01	16.0 ± 0.10
*Listeria monocytogenes* ATCC 19114	9.0 ^b^ ± 0.04	11.0 ^a^ ± 0.01
*Micrococcus luteus* ATCC 10240	23.0 ^a^ ± 0.09	17.0 ^b^ ± 0.08
*Bacillus coagulans* (laboratory isolate)	13.0 ^b^± 0.02	16.0 ^a^ ± 0.10
*Staphylococcus aureus* ATCC 29737	15.0 ^b^ ± 0.09	25.0 ^a^ ± 0.12

Values are means ± SE of triplicate samples. Means not sharing a common superscript (a or b) in the same row are significantly different at *p* ≤ 0.05.

**Table 2 antioxidants-11-01354-t002:** Cooking characteristics of beef burger formulated with different concentrations of clove powder during storage at 4 °C (±1).

Clove Powder (%)	Storage Period (Days)			
	0	7	14	21
Cooking Yield (%)				
0	85.0 ± 0.46 ^cp^	84.8 ± 0.35 ^bp^	ND	ND
2	88.9 ± 0.97 ^ap^	84.8 ± 0.79 ^bs^	87.0 ± 0.09 ^bq^	85.4 ± 0.67 ^ar^
4	86.8 ± 0.52 ^bq^	84.9 ± 0.78 ^br^	87.9 ± 0.78 ^ap^	84.8 ± 0.59 ^ar^
6	86.6 ± 0.65 ^br^	90.9 ± 1.01 ^ap^	87.8 ± 0.71 ^aq^	84.7 ± 1.02 ^as^
Fat retention (%)				
0	86.5 ± 0.21 ^bq^	88.5 ± 0.24 ^dp^	ND	ND
2	88.0 ± 0.69 ^as^	92.6 ± 0.93 ^ap^	88.3 ± 0.07 ^ar^	89.4 ± 0.38 ^aq^
4	87.5 ± 0.13 ^ar^	90.3 ± 0.32 ^bp^	87.9 ± 0.28 ^br^	88.2 ± 0.28 ^bq^
6	87.5 ± 0.53 ^aq^	89.3 ± 0.27 ^cp^	87.9 ± 0.14 ^bq^	87.9 ± 0.17 ^cq^
Moisture retention (%)				
0	72.0 ± 0.83 ^cq^	74.0 ± 0.18 ^cp^	ND	ND
2	79.3 ± 0.24 ^ap^	75.3 ± 0.79 ^br^	73.2 ± 0.49 ^bs^	76.4 ± 0.39 ^aq^
4	78.3 ± 0.53 ^bp^	75.8 ± 0.75 ^bq^	75.0 ± 0.86 ^aq^	75.6 ± 0.23 ^bq^
6	78.1 ± 0.29 ^bq^	80.7 ± 0.64 ^ap^	74.7 ± 0.68 ^ar^	74.6 ± 0.31 ^cr^
Dimensional shrinkage (%)				
0	11.1 ± 0.29 ^ap^	9.0 ± 0.37 ^aq^	ND	ND
2	10.4 ± 0.34 ^bp^	8.0 ± 0.21 ^bq^	10.1 ± 0.24 ^ap^	8.3 ± 0.56 ^bq^
4	9.2 ± 0.33 ^cp^	7.5 ± 0.17 ^cq^	9.2 ± 0.23 ^bp^	9.4 ± 0.31 ^ap^
6	8.1 ± 0.54 ^dp^	7.2 ± 0.23 ^cq^	8.6 ± 0.13 ^cp^	9.0 ± 0.13 ^bp^

Values are presented as means of triplicate samples (±SE). Means not sharing common superscript(s) a, b, c, or d in a column or p, q, r, or s in a row are significantly different at *p* ≤ 0.05, as assessed by Duncan’s multiple range test. ND: not determined (spoiled).

**Table 4 antioxidants-11-01354-t004:** Color characteristics of raw beef burger formulated with different concentrations of clove powder during storage at 4 °C (±1).

Storage Period (Days)	Clove Powder (%)			
	0	2	4	6
*Lightness (L*)*				
0	50.0 ± 0.34 ^ap^	48.5 ± 0.47 ^aq^	48.9 ±0.44 ^aq^	48.0 ± 0.79 ^aq^
7	46.3 ± 0.14 ^br^	46.0 ± 0.29 ^bp^	45.4 ± 0.25 ^bq^	44.6 ± 0.88 ^bq^
14	ND	46.7 ± 0.49 ^bp^	42.8 ± 0.58 ^cq^	41.0 ± 0.26 ^cr^
21	ND	45.0 ± 0.37 ^cp^	41.5 ± 0.31 ^dq^	39.9 ± 0.58 ^dr^
*Redness (a*)*				
0	6.4 ± 0.61 ^ar^	7.0 ± 0.36 ^bq^	7.7 ± 0.27 ^ap^	6.7 ± 0.37 ^bqr^
7	6.5 ± 0.43 ^aq^	8.1 ± 0.39 ^ap^	7.9 ± 0.46 ^ap^	7.4 ± 0.74 ^ap^
14	ND	6.3 ± 0.43 ^bp^	6.1 ± 0.66 ^bp^	6.1 ± 0.71 ^bp^
21	ND	6.2 ± 0.63 ^bp^	6.1 ± 0.45 ^bp^	6.0 ± 0.68 ^bp^
*Yellowness (b*)*				
0	16.6 ± 0.46 ^aq^	17.4 ± 0.26 ^ap^	17.9 ± 0.49 ^ap^	14.8 ± 0.27 ^br^
7	16.7 ± 0.52 ^ap^	15.1 ± 0.27 ^cq^	13.7 ± 0.17 ^ds^	14.5 ± 0.32 ^br^
14	ND	16.3 ± 0.38 ^bq^	16.7 ± 0.28 ^bq^	17.3 ± 0.28 ^ap^
21	ND	14.2 ± 0.41 ^dp^	14.2 ± 0.18 ^cp^	14.7 ± 0.64 ^bp^
*Chroma (C) = (a^2^ + b^2^)^0.5^*				
0	17.7 ± 0.31 ^ar^	18.8 ± 0.27 ^aq^	19.3 ± 0.53 ^ap^	16.3 ± 0.19 ^bs^
7	17.9 ± 0.31 ^ap^	17.1 ± 0.58 ^bp^	15.8 ± 0.21 ^cr^	16.3 ± 0.13 ^bq^
14	ND	17.4 ± 0.33 ^bq^	17.8 ± 0.16 ^bq^	18.3 ± 0.22 ^ap^
21	ND	15.4 ± 0.41 ^cp^	15.5 ± 0.62 ^cp^	15.9 ± 0.15 ^cp^

Values are presented as means of triplicate samples (±SE). Means not sharing common superscript(s) a, b, c, or d in a column or p, q, r, or s in a row are significantly different at *p* ≤ 0.05, as assessed by Duncan’s multiple range test. ND: not determined (spoiled).

**Table 5 antioxidants-11-01354-t005:** Sensory properties of cooked beef burger formulated with different concentrations of clove powder during storage at 4 °C (±1).

Storage Period (Days)	Clove Powder (%)			
	0	2	4	6
*Color*				
0	8.49 ± 0.25 ^ap^	8.26 ± 0.27 ^ap^	8.24 ± 0.25 ^ap^	8.25 ± 0.31 ^ap^
7	7.23 ± 0.24 ^bq^	8.33 ± 0.21 ^ap^	7.61 ± 0.15 ^bq^	6.74 ± 0.15 ^br^
14	ND	8.06 ± 0.13 ^ap^	7.86 ± 0.15 ^bp^	6.75 ± 0.41 ^bq^
21	ND	7.21 ± 0.35 ^bp^	7.34 ± 0.37 ^bp^	6.28 ± 0.47 ^bq^
*Flavor*				
0	8.41 ± 0.24 ^aq^	9.17 ± 0.36 ^ap^	9.41 ± 0.57 ^ap^	8.82 ± 0.22 ^aq^
7	7.04 ± 0.32 ^bq^	8.41 ± 0.37 ^bp^	7.64 ± 0.29 ^bq^	7.18 ± 0.62 ^bq^
14	ND	7.24 ± 0.27 ^cp^	6.87 ± 0.13 ^cq^	6.35 ± 0.41 ^bq^
21	ND	7.13 ± 0.15 ^cp^	6.67 ± 0.22 ^cq^	6.08 ± 0.32 ^bq^
*Taste*				
0	8.17 ± 0.42 ^ap^	8.47 ± 0.26 ^ap^	8.38 ± 0.27 ^ap^	8.29 ± 0.13 ^ap^
7	7.41 ± 0.35 ^ar^	8.27 ± 0.26 ^ap^	7.34 ± 0.24 ^br^	7.89 ± 0.06 ^bq^
14	ND	6.59 ± 0.22 ^cq^	7.49 ± 0.17 ^bp^	6.17 ± 0.46 ^cq^
21	ND	7.18 ± 0.18 ^bp^	7.01 ± 0.35 ^bp^	6.13 ± 0.17 ^cq^
*Texture*				
0	8.36 ± 0.24 ^ar^	9.16 ± 0.17 ^ap^	9.02 ± 0.02 ^ap^	8.89 ± 0.06 ^aq^
7	7.17 ± 0.23 ^bq^	9.01 ± 0.47 ^ap^	8.91 ± 0.03 ^bp^	7.87 ± 0.42 ^bq^
14	ND	7.51 ± 0.28 ^bp^	7.87 ± 0.38 ^cp^	6.89 ± 0.17 ^cq^
21	ND	7.65 ± 0.32 ^bp^	6.51 ± 0.58 ^dq^	6.57 ± 0.23 ^cq^
*Juiciness*				
0	8.57 ± 0.08 ^aq^	9.12 ± 0.31 ^ap^	8.76 ± 0.09 ^ap^	8.95 ± 0.15 ^ap^
7	7.14 ± 0.31 ^bp^	7.57 ± 0.56 ^bp^	7.78 ± 0.22 ^bp^	7.81 ± 0.17 ^bp^
14	ND	7.73 ± 0.23 ^bp^	7.83 ± 0.34 ^bp^	6.75 ± 0.16 ^cq^
21	ND	7.53 ± 0.36 ^bp^	7.14 ± 0.15 ^cp^	6.55 ± 0.43 ^cq^
*Overall acceptability*				
0	8.19 ± 0.15 ^ap^	8.46 ± 0.33 ^ap^	8.76 ± 0.27 ^ap^	7.84 ± 0.02 ^aq^
7	7.62 ± 0.26 ^bq^	8.32 ± 0.26 ^ap^	7.74 ± 0.16 ^bq^	7.56 ± 0.11 ^bq^
14	ND	7.45 ± 0.38 ^bp^	7.14 ± 0.79 ^bp^	6.58 ± 0.28 ^cq^
21	ND	7.24 ± 0.29 ^bp^	7.11 ± 0.59 ^bp^	6.32 ± 0.31 ^cq^

Values are presented as means of triplicate samples (±SE). Means not sharing common superscript(s) a, b, c or d in a column or p, q or r in a row are significantly different at *p* ≤ 0.05, as assessed by Duncan’s multiple range test. ND: not determined (spoiled).

## Data Availability

Data is contained within the article.

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
