# Peer review of "Clove Polyphenolic Compounds Improve the Microbiological Status, Lipid Stability, and Sensory Attributes of Beef Burgers during Cold Storage"

_antioxidants, 2022, doi:10.3390/antiox11071354_

Round 1

Reviewer 1 Report

antioxidants-1778867

Title: Cloves polyphenolic compounds improves the microbiological status, lipid stability and sensory attributes of beef burger during cold storage

The cloves powder was used in recipe of beef burgers. The cooking characteristics (cooking yield, fat retention, moisture retention and shrinkage), oxidative characteristics and sensory characteristics of products were investigated.

Although, the results are interesting, some details should be improved and explained. I would like to make some comments that authors could take into account to improve the overall quality of the manuscript.

Comments:

Table 1. The content of table presents some features of clove powder, CP (the title of table), however the cloves powder extract (CPE) was used. It will be better if phenolic compounds will be presented as mg/g of CPE (as dry matter of CPE) and the next column as mg/100 g of CP. It is not clear if antioxidant values refer to 1 g of CPE (the dry matter or solution of CPE) or 1 g of CP. The CPE was used to determine antimicrobial activity but I do not know anything about relation between CPE and raw material (1 ml of CPE correspond to 0.333 g of clove powder?).

The names of bacteria should be written in italics.

Table 2. The units should be introduced, cooking yield (%)…

Line 384: the notation 7.0 cfu/g is not correct, please write 10^7 cfu/g.

Conclusion

Generally manuscript is well written and well discussed. It presents a new possibility of application of cloves powder into food technology.

Author Response

Title: Cloves polyphenolic compounds improves the microbiological status, lipid stability and sensory attributes of beef burger during cold storage

The cloves powder was used in recipe of beef burgers. The cooking characteristics (cooking yield, fat retention, moisture retention and shrinkage), oxidative characteristics and sensory characteristics of products were investigated.

Although, the results are interesting, some details should be improved and explained. I would like to make some comments that authors could take into account to improve the overall quality of the manuscript.

The authors thank the Reviewer for their comments and suggestions.

Comments:

Table 1. The content of table presents some features of clove powder, CP (the title of table), however the cloves powder extract (CPE) was used. It will be better if phenolic compounds will be presented as mg/g of CPE (as dry matter of CPE) and the next column as mg/100 g of CP. It is not clear if antioxidant values refer to 1 g of CPE (the dry matter or solution of CPE) or 1 g of CP.

Done. The results for CP and CPE have been added to table 1. The antioxidant activities were based on the CPE and this has been cleared in table 1 now.

The CPE was used to determine antimicrobial activity but I do not know anything about relation between CPE and raw material (1 ml of CPE correspond to 0.333 g of clove powder?).

As stated in line 90, the samples were freeze-dried and thus all the results are reported per g.

The names of bacteria should be written in italics.

Done. All the Latin names have been italicized

Table 2. The units should be introduced, cooking yield (%)…

Done. The units have been added to the table and the M&M section

Line 384: the notation 7.0 cfu/g is not correct, please write 10^7 cfu/g.

Done. The value has been changed to 7.0 log CFU/g to keep the format consistent through the discussion section.

Conclusion

Generally manuscript is well written and well discussed. It presents a new possibility of application of cloves powder into food technology.

We thank the reviewer for their positive comments.

Reviewer 2 Report

The authors propose a study on the use of clove extract in beef burgers to evaluate their shelf life, organoleptic aspects and biochemical parameters conferred by the antioxidants present.

At various concentrations, the addition of clove powder gave greater resistance to microbial agents, higher content of phytochemicals and good acceptability to sensory tests.

Research on substances that can implement the characteristics of shelf life and acceptability of foods is not a topic limited only to beef products but also vegetable ones.

The manuscript is very extensive and explores various relevant aspects with ample space for literature on the topic. However, there are some doubts about the organisation of the manuscript:

- Regarding the characteristics of the extract with methanol, it is not known whether there has been a process of concentration of the phytocompounds present in the clove powder. This makes it difficult to compare the characteristics with other literature data

- When describing the percentages of clove powder, it should be specified that it is a percentage W/W.

- Judging by the quantities of antioxidants extracted and the antioxidant effect detected, compared with the bibliographical references, it would seem that the extraction method used has reduced the activity of the phytocompounds present. It's correct it

- Paragraph 2.12 should be better developed as it does not provide any information. This is in contrast to other parts far too detailed in the methods

- In paragraph 2.17 the language seems more related to a clinical trial. It would be more correct to replace the term "experiments" with "specimens’ evaluations" and "treatments" with "formulations".

- It would be more correct to separate the results from the discussion. The presentation of these in bulk does not allow the reader to make a critical assessment of the results presented and to consider the authors' interpretation at a later time.

- Is it correct to compare the results of the flavonoid content (mg CE/g) in the manuscript with that obtained from other groups (mg QE/g)?

- The authors associate the variability of the results in the literature with cultivar differences and other objective aspects but it is evident that there are many subjective differences regarding the type of extraction. This aspect should be emphasized more

- Paragraphs 3.3 to 3.8 appear very generalized. For example, lines 486-487, and other passages in the paragraphs indicated, are not sufficiently consistent with the results. There is the impression that the authors tend to over-interpret the results. In the tables, it is necessary to add the significance (p-trend) for the storage time and cloves contents for a more rigorous interpretation

- As suggested in lines 355-358, other macromolecules may play a role associated with cloves. Why was the nutrient composition not analyzed?

- The most annoying thing is the lack of control samples at times 14 and 21 which negatively affects the robustness of the results, often over-interpreted. Understandably, they are not used in organoleptic tests but there was no reason not to be analyzed and reported in the other evaluations. For example, It turns out in the text that the samples were damaged due to the high microbial load but this charge is not shown in the relevant table (table 3, plate count).

Minor aspects:

- Some extra spaces on lines 43, 235, 344, 423

- On line 12, the verb "was" seems out of place

- Line 70 seems to be missing an "as" before "a natural"

- In line 70 between "current" and "was" the term "study" seems to be missing

- After the first abbreviation of cloves powder, the acronym should always be used

- On line 119 "the formulations consist of 4 concentrations of CP"

- On line 234 the term “also” seems out of place

- QE/g must be defined the first time it is used

- The header of table 1 should specify "extract"

- Where is paragraph 3.6?

Author Response

Reviewer 2

Comments and Suggestions for Authors

The authors propose a study on the use of clove extract in beef burgers to evaluate their shelf life, organoleptic aspects and biochemical parameters conferred by the antioxidants present.

At various concentrations, the addition of clove powder gave greater resistance to microbial agents, higher content of phytochemicals and good acceptability to sensory tests.

Research on substances that can implement the characteristics of shelf life and acceptability of foods is not a topic limited only to beef products but also vegetable ones.

The manuscript is very extensive and explores various relevant aspects with ample space for literature on the topic. However, there are some doubts about the organisation of the manuscript:

The authors thank the reviewer for their positive comments.

- Regarding the characteristics of the extract with methanol, it is not known whether there has been a process of concentration of the phytocompounds present in the clove powder. This makes it difficult to compare the characteristics with other literature data.

- When describing the percentages of clove powder, it should be specified that it is a percentage W/W.

The use of 2%, 4% and 6% is self-explanatory, and avoid the use of “g/ 100g sample” term, which will be more troublesome to repeat use.

- Judging by the quantities of antioxidants extracted and the antioxidant effect detected, compared with the bibliographical references, it would seem that the extraction method used has reduced the activity of the phytocompounds present. It's correct it

It is the opposite. As shown and discussed in L240-252 (shown below), the values in the present study are higher than those reported in the literature.

“Ali et al. (2021) extracted bioactive compounds from clove powder using 70% ethanol in MilliQ water containing 0-1% formic acid and reported TPC and TFC contents of 215.1 mg GAE/g and 5.6 mg QE/g, respectively. Frohlich et al. [35] applied an ultrasound-assisted extraction method using aqueous ethanol (70%, at 60°C, for 20 min) as a solvent for extracting bioactive compounds from clove leaves and found a TPC of 387.9 mg GAE/g. El-Maati et al. [19] used three different solvents (water, ethanol, and ethyl acetate) for the extraction of phenolic compounds from clove bud and observed that water is the best solvent yielding TPC, TFC and DPPH radical scavenging values of 230 mg GAE/g extract, 17.5 mg QE/g extract, and 91.4%, respectively. Sultana [36] extracted bioactive compounds from clove seeds using different solvents (30% and 70% aqueous methanol and 0.5-1.0 N HCl acidified methanol) for 24 h and found that acidified methanol extracts contain high TPC (115.3 mg GAE/100 g dry matter) and DPPH radical scavenging activity (94.6%).  The results of TPC and TFC in the aforementioned studies were lower than those found in the present study”

- Paragraph 2.12 should be better developed as it does not provide any information. This is in contrast to other parts far too detailed in the methods

Thank you for this suggestion. The equations used have been added to the text.

- In paragraph 2.17 the language seems more related to a clinical trial. It would be more correct to replace the term "experiments" with "specimens’ evaluations" and "treatments" with "formulations".

The language used is appropriate for academic and research contexts. “experiments”  and “treatments” are the standard terms used to describe research and they are not limited to the medical field.

- It would be more correct to separate the results from the discussion. The presentation of these in bulk does not allow the reader to make a critical assessment of the results presented and to consider the authors' interpretation at a later time.

The authors prefer to keep the results and discussion in a combined section to avoid repetition (it is already 12 pages and to increase the text by splitting the results and discussion would only make the MS longer. The authors believe the current format is a coherent presentation. The complete tables and figures should provide the full picture of the findings that enable the readers to make the critical assessments required. Having the results and discussion in one section is useful to have the key results and the interpretation side by side.

- Is it correct to compare the results of the flavonoid content (mg CE/g) in the manuscript with that obtained from other groups (mg QE/g)?

We agree with the reviewer that direct comparison is not appropriate since the standards are different. We added a new reference that reported the TFC based on CE/g extract but used different solvent extraction. The variations have been attributed “The variations in the bioactive properties (TPC, TFC, and DPPH inhibition) among these studies could be attributed to the differences in the genotypes, parts, maturity stages, cultivation location and seasons, postharvest processing conditions of cloves and the extraction conditions (extraction method, solvent type, polarity, sample/solvent ratio, temperature, and duration) of bioactive compounds as well as the reference standards used in the assays.”

- The authors associate the variability of the results in the literature with cultivar differences and other objective aspects but it is evident that there are many subjective differences regarding the type of extraction. This aspect should be emphasized more

. The variations have been attributed “The variations in the bioactive properties (TPC, TFC, and DPPH inhibition) among these studies could be attributed to the differences in the genotypes, parts, maturity stages, cultivation location and seasons, postharvest processing conditions of cloves and the extraction conditions (extraction method, solvent type, polarity, sample/solvent ratio, temperature, and duration) of bioactive compounds as well as the reference standards used in the assays.”

This study is not about extensive characterization of clove extracts. We aimed to benchmark the extract used in the present meat study by providing sufficient information about the composition and the bioactive contents present in the extract used. An extensive comparison of various parameters used in eth extraction with those reported in the literature is not appropriate for this study.

- Paragraphs 3.3 to 3.8 appear very generalized. For example, lines 486-487, and other passages in the paragraphs indicated, are not sufficiently consistent with the results. There is the impression that the authors tend to over-interpret the results.

The authors are not clear about this comment and are not sure what the reviewer is advising.

In the tables, it is necessary to add the significance (p-trend) for the storage time and cloves contents for a more rigorous interpretation

Adding P-value for each raw and P-value for each measured parameter in each column is going to create messy tables. We believe the standard format of adding superscripts with a footnote “Means not sharing a common superscript(s) a, b, or c in a column or p or q in a row are significantly different at P ≤ 0.05 as assessed by Duncan's multiple range test.” Will be a more clear presentation that matches standard published studies.

- As suggested in lines 355-358, other macromolecules may play a role associated with cloves. Why was the nutrient composition not analyzed?

To calculate the cooking properties, the fat and moisture contents have been determined (please see the equations added in section 2.12) and have been used in the determination of the presented cooking properties. Therefore, the authors felt that there is no need to repeat the presentation of the results as reviewers do not appreciate double-dipping.

- The most annoying thing is the lack of control samples at times 14 and 21 which negatively affects the robustness of the results, often over-interpreted. Understandably, they are not used in organoleptic tests but there was no reason not to be analyzed and reported in the other evaluations. For example, It turns out in the text that the samples were damaged due to the high microbial load but this charge is not shown in the relevant table (table 3, plate count).

Once the samples are not suitable for consumption, any other information is longer of value. It will be just a waste of resources for no gain.

Minor aspects:

- Some extra spaces on lines 43, 235, 344, 423

Thank you. The extra spaces have been removed

- On line 12, the verb "was" seems out of place

Thank you. The word “was” has been removed

- Line 70 seems to be missing an "as" before "a natural"

Thank you. The word “as” has been added

- In line 70 between "current" and "was" the term "study" seems to be missing

Thank you. The word “study” has been added

- After the first abbreviation of cloves powder, the acronym should always be used

Done. The abbreviation is used consistently when we refer to extracts from this study. For those reported in the literature, we used the reported terminology

- On line 119 "the formulations consist of 4 concentrations of CP"

Done. Thank you.

- On line 234 the term “also” seems out of place

Thank you. The word “was” has been removed

- QE/g must be defined the first time it is used

Thank you. The word “QE” has been explained

- The header of table 1 should specify "extract"

The title has been modified

- Where is paragraph 3.6?

Thank you. The number of sections has been fixed

Round 2

Reviewer 2 Report

Despite much resistance, the authors have sufficiently responded to the requests.

In the case of some aspects, however relevant, the authors preferred to stay in their positions. I rely on the editor's judgment, should these aspects be decisive.